# Restoring Degraded Landscapes through an Integrated Approach Using Geospatial Technologies in the Context of the Humanitarian Crisis in Cox's Bazar, Bangladesh

**Rashed Jalal** [1,*], **Rajib Mahamud** [1], **Md. Tanjimul Alam Arif** [1], **Saimunnahar Ritu** [1], **Mondal Falgoonee Kumar** [1], **Bayes Ahmed** [2], **Md. Humayun Kabir** [3], **Mohammad Sohal Rana** [3], **Howlader Nazmul Huda** [4], **Marco DeGaetano** [1], **Peter John Agnew** [1], **Amit Ghosh** [1], **Fatima Mushtaq** [1], **Pablo Martín-Ortega** [1], **Andreas Vollrath** [1], **Yelena Finegold** [1], **Gianluca Franceschini** [1], **Rémi d'Annunzio** [1], **Inge Jonckheere** [1] and **Matieu Henry** [1]

1   Food and Agriculture Organization of the United Nations, 00153 Rome, Italy
2   Institute for Risk and Disaster Reduction (IRDR), University College London (UCL), Gower Street, London WC1E 6BT, UK
3   Bangladesh Forest Department, Ministry of Environment Forest and Climate Change, Sherebangla Nagar, Agargaon, Dhaka 1207, Bangladesh
4   International Organization for Migration, House 13A, Road 136, Gulshan 1, Dhaka 1212, Bangladesh
*   Correspondence: rashed.jalal@fao.org

**Abstract:** The influx of nearly a million refugees from Myanmar's Rakhine state to Cox's Bazar, Bangladesh, in August 2017 put significant pressure on the regional landscape leading to land degradation due to biomass removal to provide shelter and fuel energy and posed critical challenges for both host and displaced population. This article emphasizes geospatial applications at different stages of addressing land degradation in Cox's Bazar. A wide range of data and methods were used to delineate land tenure, estimate wood fuel demand and supply, assess land degradation, evaluate land restoration suitability, and monitor restoration activities. The quantitative and spatially explicit information from these geospatial assessments integrated with the technical guidelines for sustainable land management and an adaptive management strategy was critical in enabling a collaborative, multi-disciplinary and evidence-based approach to successfully restoring degraded landscapes in a displacement setting.

**Keywords:** land degradation; sustainable land management; emergency; earth observation; sustainable development goals; Rohingya

## 1. Introduction

Productive land is the foundation of global food security and environmental health, zero hunger, poverty eradication and energy for all. However, this finite resource is under continuous threat. Anthropic activities affects more than 70% of the global ice-free land surface [1]. Globally, the biophysical status of 5670 million hectares (ha) of land is declining, of which 1660 million ha (29%) is attributed to human-induced land degradation [2]. With up to 40% of the planet's land degraded, which would reach the size of South America by 2050, land degradation is recognized as one of the significant environmental threats to society, directly affecting half of humanity and threatening roughly half of the global gross domestic product (GDP) [3]. Addressing land degradation through sustainable management of natural resources and socioeconomic development, as well as "strengthen cooperation on desertification, dust storms, land degradation and drought and promote resilience and disaster risk reduction", is recognized in the 2030 Agenda for Sustainable Development [4].

Land degradation is intrinsically related to other environmental challenges, including climate change, loss of biodiversity, and humanitarian crises. In 2020, more than 70 million

forcibly displaced persons were scattered globally [5], and the number continues to grow. Since 2008, climate refugees have been growing by more than 20 million people annually [6] and could reach 140–200 million people by 2050 [7]. This expanding humanitarian crisis potentially affects every people and ecosystem, putting refugees as both a cause and victim of environmental and land degradation [8]. Restoring degraded land in and around refugee camps can bring positive externalities to displaced populations and host communities, and is increasingly being taken into consideration given the associated environmental challenges affecting humanitarian settings.

In Bangladesh, one of the most vulnerable countries to climate change [9], the sudden influx of nearly a million Rohingya refugees/Forcibly Displaced Myanmar Nationals (FDMNs) from the Rakhine state, Myanmar in August 2017, made Cox's Bazar (the southernmost coastal hill district of Bangladesh) home to one of the largest refugee settlements in the world. The August 2017 influx, the largest and fastest refugee influx into Bangladesh, has put substantial additional pressure on natural resources and increased already existing challenges to human health, food security, nutrition, water supply and sanitation, shelter, education, access to energy and environmental services, not only for the people displaced but also for their host communities. To address this situation, the major stakeholders have made a joint effort to rehabilitate the degraded landscapes inside and outside the camp area since 2018. As a result, a total of 450 ha (approximately) of degraded areas were brought under different restoration activities by different agencies, and an additional 2000 ha of degraded forestland was maintained jointly with the Bangladesh Forest Department (BFD) of the Government of Bangladesh (GoB).

Assessment and monitoring of restoration projects are often complex due to the challenges related to accessibility, lack of affordable and appropriate methodologies, difficulty in obtaining long-term data and lack of funds, together with capacity constraints in general [10]. Advancements in geospatial and earth observation technology and the availability of higher resolution satellite data have considerable potential in effectively delivering timely, cost-effective, reliable, and homogeneous information. However, only some examples of the use of geospatial technologies to assess restoration interventions are available [11]. In general, there is a lack of evaluation and dissemination of the restoration results, representing a constraint on applying the best technologies and approaches available [12]. There is also a broad consensus on the need for innovative approaches to systematically evaluate the effectiveness of restoration efforts [10,12,13].

In this context, this article aims to present the ongoing geospatial approach that integrated and facilitated different aspects of addressing land degradation in a displacement setting in Cox's Bazar, Bangladesh. The overall approach to land degradation assessment and implementation of restoration activities evolved over time, taking into consideration the availability of data and methods, need, capacity, expertise, and lessons learned. The approach included delineating forest land boundaries, land cover mapping, wood fuel supply and demand assessment, land degradation assessment, preparation of technical specifications and suitability analysis for restoration activities, and implementation and monitoring of restoration activities.

## 2. Materials and Methods

A wide range of data was collected or prepared using different methods to support the process of addressing land degradation. Table 1 summarizes the list of data used for various applications. After a brief overview of the study area, descriptions of the methods are provided in subsequent sub-sections.

**Table 1.** Data used in different stages of addressing land degradation.

| Data | Accessed from [1] | Date of Data | Application |
|---|---|---|---|
| **Areas of interest** | | | |
| Rohingya refugee camps | HDX | 2018 | Area of interest |
| Cox's Bazar south forest division | BFD | 1920s (CS sheets), updated and digitized in 2018 | Area of interest |
| Forest land boundaries | BFD | 1920s (CS sheets), updated and digitized in 2018 | Area of interest and land suitability assessment |
| **Satellite image** | | | |
| Sentinel 2 images | GEE | 2017 to 2019 | Land degradation assessment |
| Landsat 4, 5 and 8 images | GEE | 2003 to 2021 | Restoration monitoring |
| **Other** | | | |
| Buildings, roads, water body footprints | HDX | 2019 | Land suitability assessment (inside the camps) |
| Protected areas | BFD | 2018 | Land suitability assessment |
| Land cover 2015 | BFD | 2015 | Land suitability assessment |
| Digital elevation model (0.5 m resolution) | IOM-NPM | 2019 | Land suitability assessment (inside the camps) |
| SRTM Digital elevation model (30 m resolution) | GEE | 2000 | Land suitability assessment (outside the camps) |
| **Elephant path** | BFD | 2016 | Land suitability assessment |
| Restoration activity areas | FAO | 2018 and 2019 | Restoration monitoring |
| Wood fuel supply and demand | FAO-IOM | 2016 and 2017 | Restoration planning |

[1] BFD—Bangladesh Forest Department; CS—Cadastral survey; FAO—Food and Agriculture Organization of the United Nations; GEE—Google Earth Engine; HDX—Humanitarian Data Exchange; IOM—International Organization for Migration; NPM—Needs and Population Monitoring.

## 2.1. Study Area and Context

Cox's Bazar district, with an area of about 2492 km$^2$, is located between 20°43′ and 21°56′ north latitudes and 91°50′ and 92°23′ east longitudes. The district is located at the fringe of the Bay of Bengal with an unbroken sea beach, the longest one in the world. It is bounded by Chattogram district to the north, Bandarban district and Myanmar to the east, and the Bay of Bengal to the south and west (Figure 1).

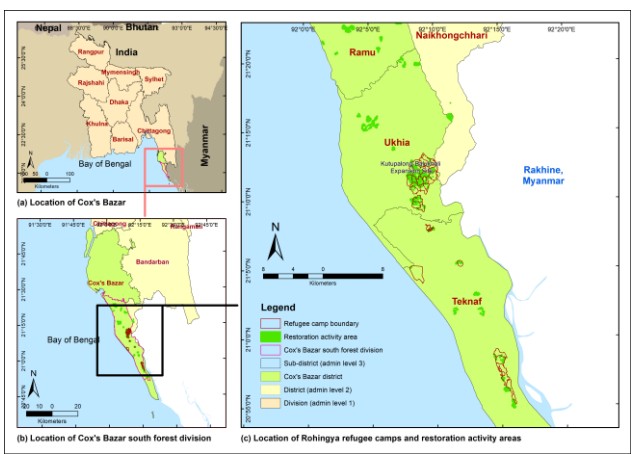

**Figure 1.** Location of Cox's Bazar district (**a**), Cox's Bazar south forest division (**b**), refugee camps and areas of restoration activities (**c**).

As of October 2022, over 943,000 Rohingya refugees/FDMNs reside in the Ukhiya and Teknaf sub-districts [14] in Cox's Bazar south forest division, an administrative area for the management of forest land by the BFD. The Cox's Bazar south forest division covers a significant part of the hill forests of the country, representing features of tropical evergreen and semi-evergreen forests, and has one of the most species-rich and productive reserve forests. However, these natural resources are becoming degraded through illegal logging, encroachment, hill-cutting, forest fires, shifting cultivation, human settlement, agriculture and horticulture expansion, and clear-felling followed by commercial plantation with short rotation of exotic species [15,16].

Such prevailing land degradation dynamics in the area are further exacerbated by fluctuating but persistent arrivals of Rohingya refugees/FDMNs, with a massive influx of around 742,000 Rohingya refugees/FDMNs since 25 August 2017 [5]. The vast majority live in 34 extremely congested refugee camps, including the largest single site, the Kutupalong-Balukhali Expansion Site, which accommodates more than 635,000 Rohingya refugees/FDMNs [14]. All the camps are in Cox's Bazar south forest division. This put significant pressure on the regional landscape resulting from removal of trees roots and cover grass to provide shelter and fuel for this forcefully displaced population. Figure 2 depicts the loss of vegetation from February 2017 to February 2018 due to the expansion of the Kutupalong-Balukhali refugee camp.

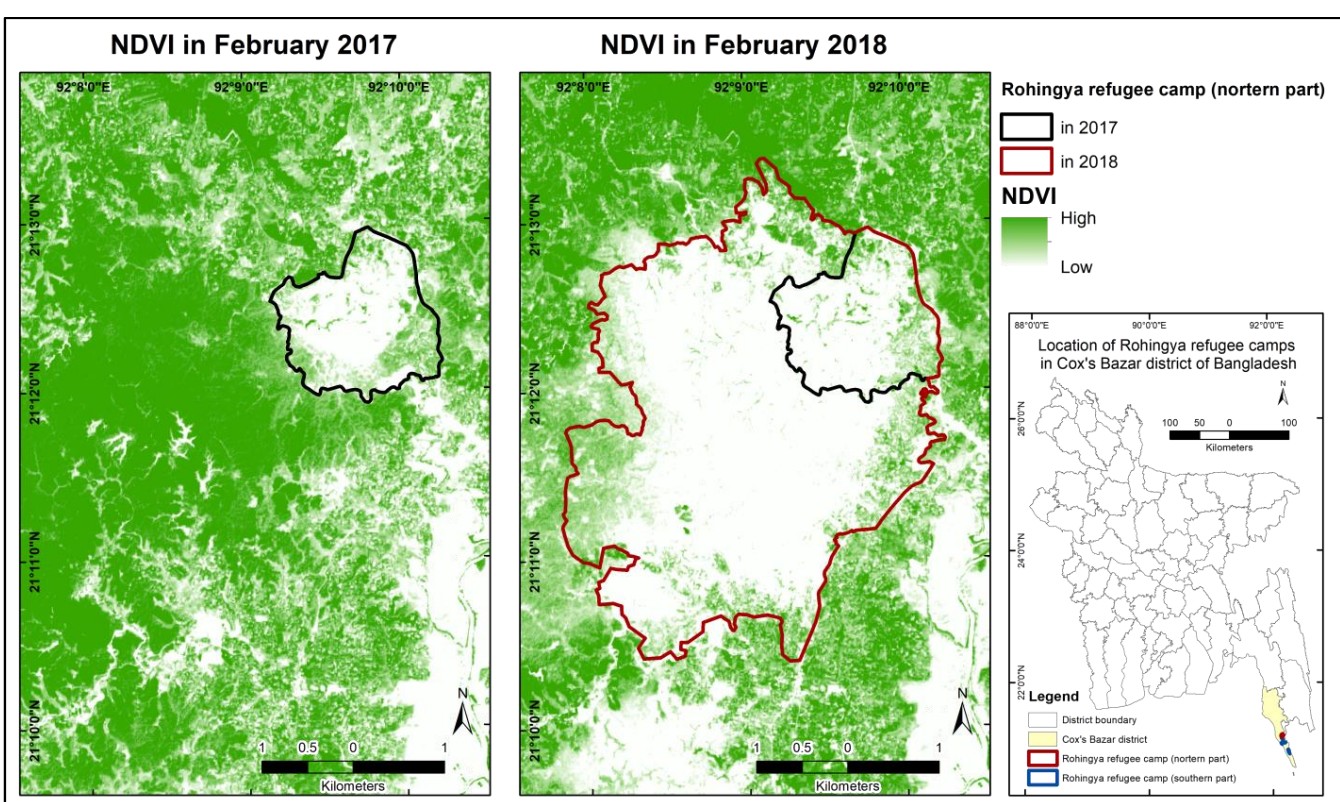

**Figure 2.** Change of vegetation between 2017 and 2018 as depicted by a decreased normalized difference vegetation index (NDVI). A lower NDVI means less vegetation cover.

In response, various national and international agencies coordinated by the Energy and Environment Technical Working Group (EETWG), in close collaboration with the BFD and the local host communities and Rohingya refugees/FDMNs, have been working together to implement an integrated land restoration approach to rehabilitate the degraded lands inside and outside the camp area since 2018. This article considered restoration activities in about 531 ha of land for which geographic boundaries were available. Table 2 presents the distribution of different restoration activities' areas, times, and locations.

**Table 2.** Area distribution and location of different restoration activities.

| Type of Activity | Activity Started | Location | Area (ha) |
|---|---|---|---|
| Forest restoration | 2018 | Outside refugee camp | 0.36 |
| | | Inside refugee camp | 27.51 |
| | 2019 | Outside refugee camp | 298.23 |
| | | Inside refugee camp | 66.53 |
| Land stabilization | 2018 | Inside refugee camp | 9.33 |
| | 2019 | Outside refugee camp | 0.78 |
| | | Inside refugee camp | 4.96 |
| Reforestation | 2018 | Outside refugee camp | 11.36 |
| | | Inside refugee camp | 33.97 |
| | 2019 | Outside refugee camp | 38.08 |
| | | Inside refugee camp | 40.12 |
| | | **Total** | **531.23** |

## 2.2. Delineation of Forest Land Boundaries and Land Cover Mapping

The forest land boundaries of Cox's Bazar south forest division were delineated from available cadastral survey (CS) sheets. The CS sheets were scanned and geo-referenced using differential global positioning systems (DGPS). The geo-referenced images were digitized to prepare the GIS layer and were further corrected using IKONOS (acquired in 2012), RapidEye (acquired in 2012) and IRS PAN (acquired in 2004) satellite images of the Cox's Bazar south forest division available in the BFD archive. The edges of each sheet map were matched with adjacent ones, and positional accuracy was compared with reference points collected from the field using real-time kinematic (RTK) positioning [17].

The 2015 national land cover map [18] was used as the baseline land information. It was developed using multi-spectral ortho (Level 3) SPOT6/7 four-band images of 6-m spatial resolution with a maximum of 10% cloud coverage. An object-based image analysis (OBIA) approach was adopted to create image objects, followed by a visual image interpretation technique to classify land cover. The overall accuracy of the 2015 national land cover map was estimated at 89% [19].

## 2.3. Wood Fuel Supply and Demand Assessment

An assessment of wood fuel supply and demand was conducted in 2016 and was later updated in 2017 [20] after the August 2017 influx. The assessment combined field and remote sensing data following the recommended approach by d'Annunzio, R. et al. [21] for assessing wood fuel supply and demand in displacement settings. The process included an assessment of standing woody biomass available for use as fuel (fuel wood supply), the changes they had undergone over a given period, consumption over the same time (assuming wood fuel consumption is equal to wood fuel demand) and the gap between demand and supply.

The assessment of supply was performed by combining field measurements for aboveground biomass stock with land cover changes based on historical satellite image time series analysis. For assessing the biomass stock in close proximity of refugee camps, samples were taken randomly from different land covers (based on the 2015 land cover map), having the potential for supplying wood fuel. A total of 15 plots were measured. The plot design and wood fuel assessment followed the same procedures as in the Bangladesh Forest Inventory [22], where each sample plot consists of 5 sub-plots (1 in the center and the remaining four in four cardinal directions), each with a radius of 19 meters. For demand

assessment, household interviews, participatory rural appraisals (PRA) and focused group discussions (FGDs) were conducted to assess the fuel wood energy consumption of varying social units inhabiting the area.

### 2.4. Land Degradation Assessment

Land degradation mapping was performed employing a before and after land cover change analysis using Sentinel 2 multispectral 10 m images for February 2017 and 2018. Five broad land cover classes (i.e., water, settlement, bare land, sparse vegetation and dense vegetation) were delineated for 2017 and 2018 based on NDVI thresholds defined through expert judgements. Different levels of land degradation were identified throughout Cox's Bazar south forest division based on the land cover change from 2017 to 2018 as follows:

- High degradation: if the dense vegetation class in February 2017 was converted to bare land, settlement or water in February 2018.
- Medium degradation: if the sparse vegetation class in February 2017 was converted to bare land, settlement or water in February 2018.
- Low degradation: if the dense vegetation class in February 2017 was converted to sparse vegetation in February 2018.

### 2.5. Technical Specifications for Restoration Activities

Technical specifications for the restoration activities were prepared using the World Overview of Conservation Approaches and Technologies (WOCAT) and documented in a living report [23] in consultation with experts and stakeholders. Particular attention was paid to the sustainability of plantation activities, considering aspects such as the supply of seedlings from nurseries, vegetation growth, vegetation layers, inputs and workforce, as well as ensuring the maintenance of plant biodiversity and promoting the use of native species.

### 2.6. Suitability Analysis for Restoration Activities

Various spatial data (e.g., land cover, forest land boundary, slope, altitude, roads, river, elephant path, flood risk due to low elevation and protected areas) were integrated with land degradation to perform the land suitability assessment based on criteria identified in consultation with local and national experts to identify potential and priority areas for restoration. The process is continuously being updated as more data become available.

### 2.7. Implementation of Restoration Activities

A collaborative process was established by the Energy Environmental Technical Working Group (EETWG) and the Inter Sector Coordination Group (ISCG) to support the coordination, planning and implementation of restoration activities inside the camps. The site management and site development (SMSD) team assisted in overall camp planning and management. The BFD played a crucial role in providing technical guidance in the design and implementation process (e.g., plant selection, plantation management, and logistics) of restoration activities inside and outside the camps.

In coordination with EETWG, implementing partner organizations mapped the available areas for restoration interventions in respective camps. Field area mapping was conducted using GPS. Plantation targets were set depending on the budget and human resources available. Specific areas were allocated for each organization, and documents (including maps) were maintained to avoid overlapping and gaps. The degradation map and potential restoration areas were verified on the ground jointly by relevant stakeholders and implementing agencies and approved by the authorized government departments before the initiation of restoration works. Considering the importance of quality planting materials for the success of any restoration initiative, plant nurseries were developed with support from the BFD and the host communities around the camps.

*2.8. Monitoring Restoration Activities*

The productivity state, one of the three metrics for calculating the land productivity sub-indicator for the sustainable development goals (SDG) indicator 15.3.1—Proportion of land that is degraded over total land area [24], was used to assess the performance of restoration activities. The productivity state in the monitoring periods can be calculated from the 16 most recent years of annual net primary productivity (NPP) of vegetation data up to and including the most recent year in the monitoring period. The mean of the most recent three years is compared to the distribution of annual NPP values in the preceding 13 years. For calculating the land productivity sub-indicator and reporting on the SDG indicator 15.3.1, the good practice guidance for the SDG indicator 15.3.1 [24] recommended that only the areas of the lowest negative $Z$ score ($<-1.96$) be considered as degraded and other areas as not degraded.

In this study, the productivity states of two monitoring periods of 2019 to 2021 and 2016 to 2018 were calculated by comparing the mean annual normalized difference vegetation index (NDVI), as a proxy of NPP, to the distribution of annual NDVI values observed in 2003 to 2015. A fixed baseline period of 2003 to 2015 was used to facilitate a direct comparison of the $Z$ scores between the monitoring periods. Annual NDVI estimates for the restoration areas were retrieved from Landsat 4, 5 and 8. Productivity states were calculated as follows: calculate the mean ($\mu$) and standard deviation ($\sigma$) of the annual NDVI estimates from 2003 to 2015 inclusive (Equations (1) and (2), respectively), calculate the means of the yearly NDVI estimates for the monitoring periods (Equation (3)) and calculate the $Z$ statistics for the monitoring periods (Equation (4)).

$$\mu = \frac{\sum_{2003}^{2015} x}{13} \tag{1}$$

$$\sigma = \sqrt{\frac{\sum_{2003}^{2015}(x-\mu)^2}{13}} \tag{2}$$

$$\overline{x} = \frac{\sum_{y-2}^{y} x}{3} \tag{3}$$

$$z = \frac{\overline{x} - \mu}{\sigma / \sqrt{3}} \tag{4}$$

where $x$ is the annual NDVI and $y$ is the end year of the monitoring period. Pixel-wise differences of $Z$ scores between 2019 to 2021 and 2016 to 2018 (subtracting the $Z$ score of 2016 to 2018 from the $Z$ score of 2019 to 2021) were calculated. Hence, positive difference indicates improvement of state in 2019 to 2021 compared with 2016 to 2018 and vice versa. *t*-tests were conducted to assess statistical significances and effect sizes [25] of the changes and directions in productivity states considering different aspects (i.e., type, time and location) of restoration activities. $Z$ scores were calculated using the SDG 15.3.1 module of SEPAL (https://sepal.io/, accessed on 20 January 2023), and *t*-tests were performed using R.

**3. Results**

*3.1. Forest Land Delineation*

From the forest land delineation, 67,692 ha of land boundary in Cox's Bazar south forest division was demarcated, of which 42,686 ha (63%) were forest land under the management of the BFD. 35,801 ha of the forest land were identified as reserved and 6884 ha as protected forest. Of the 2639 ha of refugee camp area, about 75% was in the forest land (69% reserved forest and 6% protected forest), as shown in Figure 3.

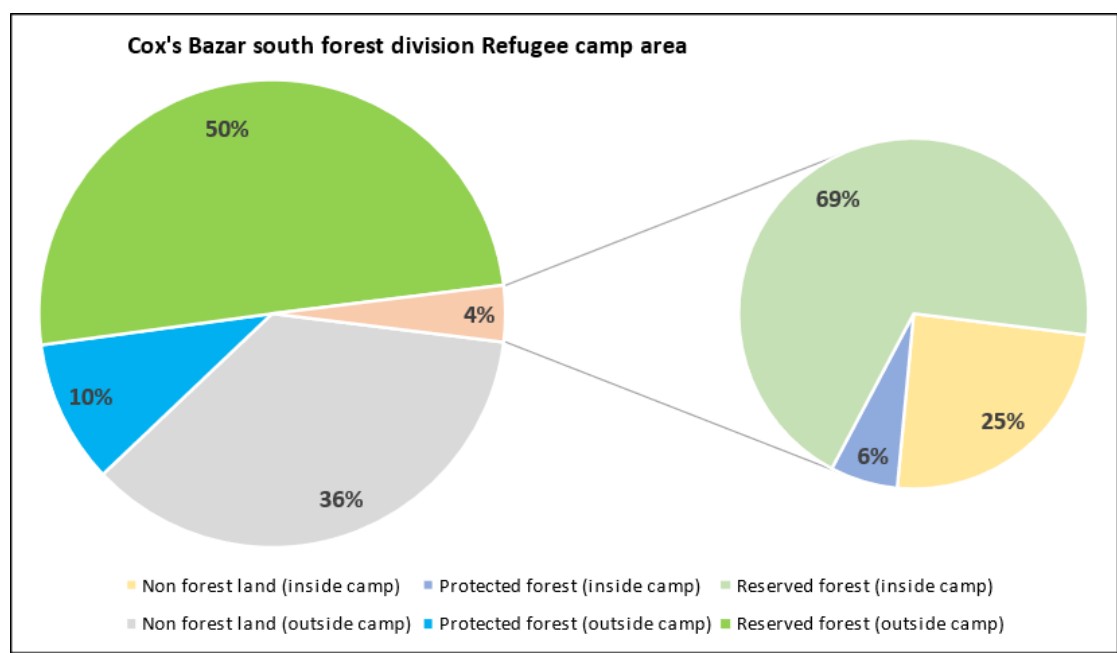

**Figure 3.** Distribution of forest land area with Cox's Bazar south forest division and refugee camp area.

### 3.2. Wood Fuel Supply and Demand

The fuelwood demand assessment revealed a six-fold increase in fuelwood demand, from 54,542 tons per year in 2016 to 312,807 tons per year in 2017 (estimated from the total number of refugee households), while the entire available stock was estimated as 331,266 tons of dry biomass. This revealed immense pressure on existing forest resources, indicating a complete loss of forestlands if the land degradation continued for a few years.

### 3.3. Land Degradation

In total, 7220 ha of land (about 11% of the total area) were degraded within one year, of which about 74% was BFD forest land (i.e., protected or reserved forest). BFD forest lands, especially within and near the refugee camp area, were highly impacted by different levels of degradation. As shown in Table 3, about 494 ha and 836 ha of land inside the camp were high and medium degraded, of which 99% and 90% were BFD forest land, respectively. Land degradation maps [26–28] were prepared and published for wider dissemination and sensitization among the stakeholders.

**Table 3.** Land degradation in Cox's Bazar south forest division (areas are in ha).

| | Forest Type | Degradation | | | Other (Enhancement or No Change) | Total |
|---|---|---|---|---|---|---|
| | | **High** | **Medium** | **Low** | | |
| Inside camp | Non forest land | 4 | 79 | 22 | 543 | 648 |
| | Protected forest | 5 | 56 | 4 | 101 | 165 |
| | Reserved forest | 484 | 701 | 25 | 616 | 1826 |
| within 1 km from camp boundary | Non forest land | 9 | 107 | 42 | 1533 | 1691 |
| | Protected forest | 2 | 32 | 8 | 220 | 262 |
| | Reserved forest | 315 | 411 | 349 | 2684 | 3760 |

**Table 3.** *Cont.*

| | Forest Type | Degradation | | | Other (Enhancement or No Change) | Total |
| --- | --- | --- | --- | --- | --- | --- |
| | | High | Medium | Low | | |
| within 1–5 km from camp boundary | Non forest land | 33 | 348 | 202 | 7742 | 8325 |
| | Protected forest | 3 | 64 | 38 | 1133 | 1237 |
| | Reserved forest | 173 | 536 | 930 | 14,484 | 16,122 |
| 5 km further from the camp boundary | Non forest land | 48 | 656 | 300 | 13,338 | 14,342 |
| | Protected forest | 12 | 200 | 166 | 4843 | 5220 |
| | Reserved forest | 38 | 279 | 542 | 13,235 | 14,094 |
| Total | | 1127 | 3468 | 2625 | 60,472 | 67,692 |

### 3.4. Technical Specifications for Restoration Activities

The technical specifications [23] were prepared in consultation with stakeholders and experts. They were followed to avoid unplanned activities, protect plant biodiversity, allocate resources efficiently and improve enabling conditions to implement landscape restoration in Cox's Bazar south forest division. The activities and technical specifications were updated over time based on experiences, lessons learned and feedback from the different national and international agencies involved in landscape restoration activities. Figure 4 illustrates a representative example of schematic technical specifications and implementation of land stabilization activity on the ground. Specifically, the example demonstrates the use of multiple vegetation layers, including long-rooted grass species to stabilize topsoil, leguminous shrubs for increasing soil fertility, bamboo as living reinforcement on vulnerable slopes and fast-growing tree species for quick vegetation cover to reduce the risk of rainfall-induced landslides.

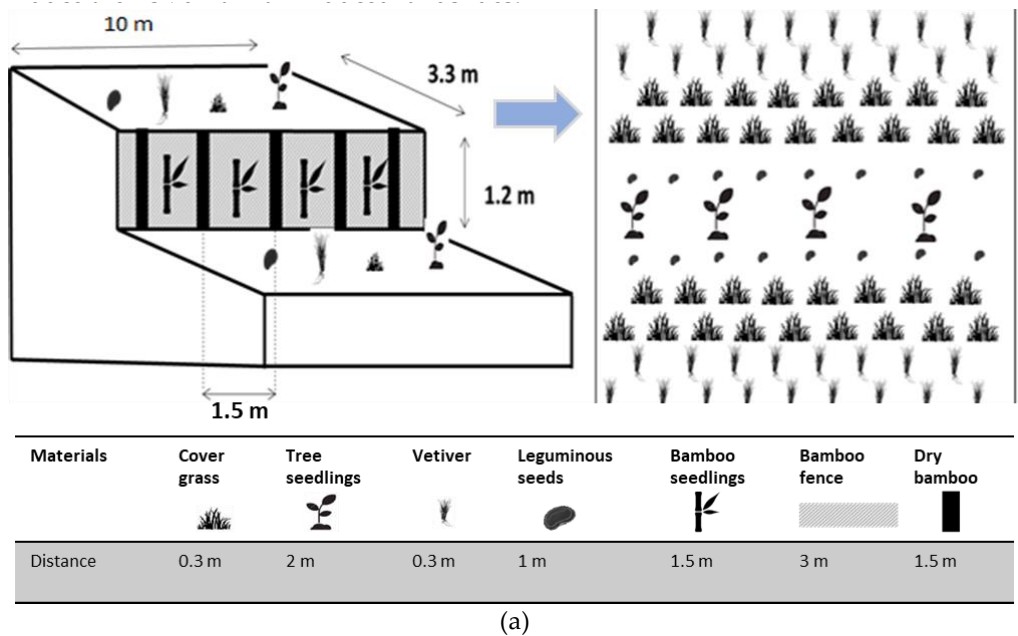

(a)

**Figure 4.** *Cont.*

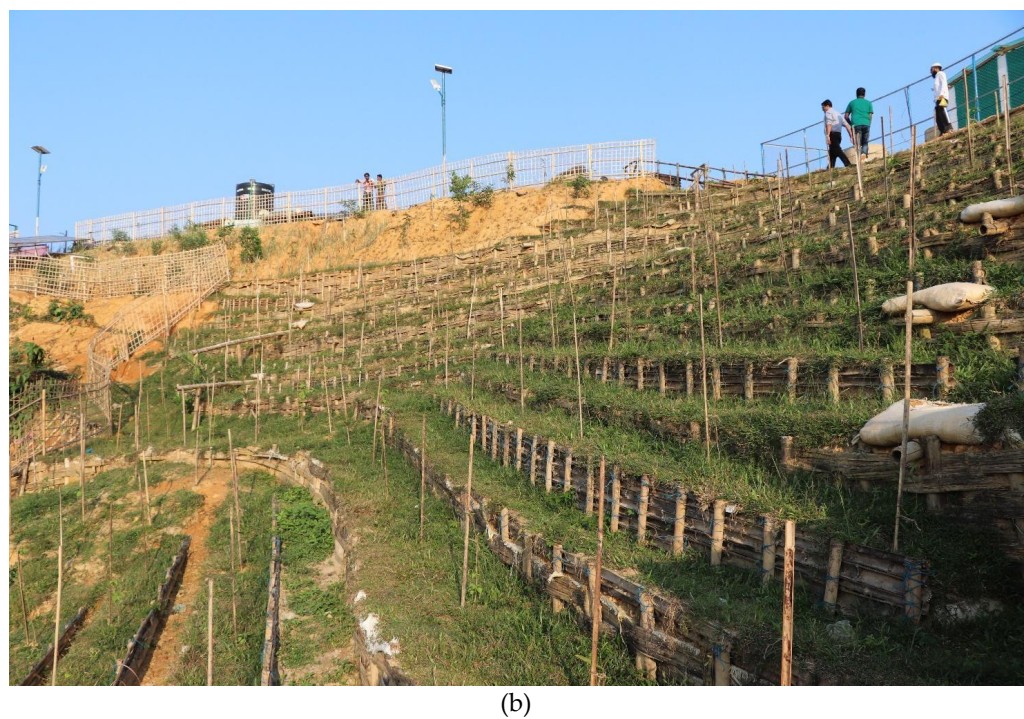

(b)

**Figure 4.** Schematic technical specification (**a**) and on-the-ground implementation of land stabilization activity (**b**) using multiple vegetation layers to reduce the risk of rainfall-induced landslides and facilitate post-landslide slope rehabilitation (photo credit: Saikat Mazumder, FAO).

*3.5. Suitability Analysis for Restoration Activities*

Suitable areas for landscape restoration were identified and published [29–31]. Depending on the emerging needs and updated data, the approach is under continuous revision. For instance, settlement footprints inside the camps and high-resolution (0.5 m) digital elevation models (DEM) were used for mapping suitable areas in 2019 inside the camp and new activities such as riparian plantation and roadside plantation were added, which were not included in the 2018 restoration plan. Table 4 presents the criteria for land suitability analysis for restoration activities inside the camps in 2019. Figure 5 shows an example map of a land restoration plan for a camp in 2019.

**Table 4.** Criteria for land suitability analysis inside the camps in 2019.

| Criteria | Restoration Activities |
|---|---|
| Bare land in January 2019 and high slope ($\geq 30°$) | Land stabilization (biological and mechanical) |
| Bare land in January 2019 and low slope ($<30°$) | Land stabilization (biological) |
| Sparse vegetation in January 2019 and non-forest in 2015 | Afforestation/reforestation |
| Sparse vegetation in January 2019 and forest in 2015 | Forest restoration |
| Dense vegetation in January 2019 | Maintenance and protection |
| Land within 5 meters from rivers and streams and within 1 meter from other water bodies | Land under use (waterside) |
| Land within 1 meter from settlements and roads | Land under use (waterside) |
| Land within 2 meters from the land under use (roadside) | Roadside plantation |
| Land within the 5 meters from the land under use (waterside) | Riparian plantation |
| Plantation in 2018 | Plantation in 2018 |

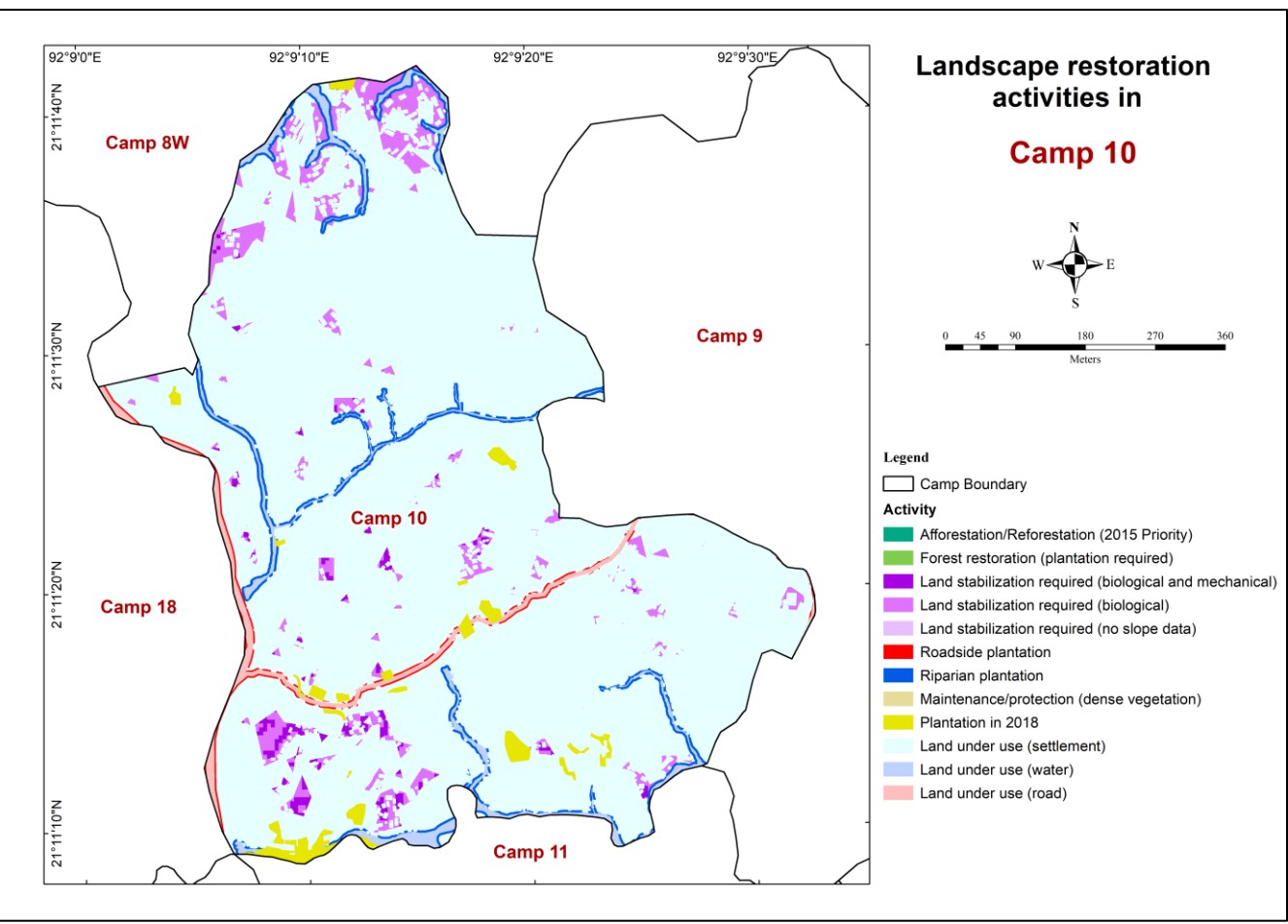

**Figure 5.** Land restoration plan map for Camp 10 in 2019.

### 3.6. Implementation of Restoration Activities

The major stakeholders have made a joint effort to engage about twenty organizations to rehabilitate the degraded forestlands inside and outside the camp area since 2018. Approximately 450 ha of degraded areas have been brought under different restoration activities by various agencies across 34 camps, involving more than 100,000 person-days. While Rohingya refugees/FDMNs were engaged inside the camps, such as in site preparation, plantation management and camp maintenance, the host communities carried out activities outside the camps, and 66 nurseries were brought into a nursery management team (NMT). An additional 2000 ha of forestland and critical watershed areas outside the camp area have been subject to restoration activities in 2020.

### 3.7. Monitoring Restoration Activities

Overall, there was a significant increase in the productivity state in 2019 to 2021 compared to 2016 to 2018, with a small effect size for all restoration areas ($t = 16.9$, $p < 0.001$, $d = 0.22$). However, mixed performances were observed for restoration areas when disaggregated by type, location and year of activity, as presented in Table 5. For restoration activities outside the camp area (about 66% of the total), significant positive change was observed with a moderate effect size ($t = 41.6$, $p < 0.001$, $d = 0.65$). However, in the remaining areas (inside the camp), a significant decrease in productivity state with a small effect size was found ($t = -15.5$, $p < 0.001$, $d = -0.35$). For the restoration activities in 2018, the productivity state decreased significantly; however, the effect size was trivial ($t = -4.8$, $p < 0.001$, $d = -0.16$). A significant and small increase in productivity state was noted for restoration works of 2019 (representing about 84% of the restoration area). Considering the type of

restoration, forest restoration and land stabilization works (representing approximately 77% of the restoration area) were found to have significant positive impacts. In contrast, there was a significant negative impact on reforestation areas with a moderate effect size (Table 5).

**Table 5.** Results from *t*-tests.

| Restoration Area | Area (ha) | df | *t* Statistic | *p* Value | Effect Size (Cohen's *d*) | Descriptor [1] |
|---|---|---|---|---|---|---|
| All | 531 | 6106 | 16.9 | $p < 0.001$ | 0.22 | Significant and small increase |
| *Location* | | | | | | |
| Inside camp | 182 | 1993 | −15.5 | $p < 0.001$ | −0.35 | Significant and small decrease |
| Outside camp | 349 | 4142 | 41.6 | $p < 0.001$ | 0.65 | Significant and moderate increase |
| *Year* | | | | | | |
| 2018 | 83 | 892 | −4.8 | $p < 0.001$ | −0.16 | Significant and trivial decrease |
| 2019 | 449 | 5225 | 21.6 | $p < 0.001$ | 0.30 | Significant and small increase |
| *Type* | | | | | | |
| Forest restoration | 393 | 4564 | 41.3 | $p < 0.001$ | 0.61 | Significant and moderate increase |
| Land stabilization | 15 | 154 | 3.5 | $p < 0.001$ | 0.28 | Significant and small increase |
| Reforestation | 124 | 1386 | −22.1 | $p < 0.001$ | −0.59 | Significant and moderate decrease |

[1] Effect size is labelled as trivial, small, moderate and large when $|d| < 0.2$, $0.2 \leq |d| < 0.5$, $0.5 \leq |d| < 0.8$, $|d| \geq 0.8$, respectively.

## 4. Discussion

There were several rapid assessments of environmental impacts following the influx in August 2017. However, they differ in methodology, data, timeframe and area of interest; hence the results are not directly comparable. For instance, one rapid environmental assessment study [32] identified about 1500 ha of forest land encroachment due to camp establishment up to November 2017. However, the assessment was semi-quantitative, and the results were not spatially explicit. Another study used satellite imagery and applied machine learning algorithms to quantity expansion of refugee settlements and estimated about 2283 ha of degradation in forest areas [33]. The study provided spatially explicit information on forest cover degradation without consideration of legal forest land boundaries.

Overall, restoration activities showed significant small to moderate improvement in at least 66% of restored areas. A global meta-analysis of 221 studies revealed an improvement of vegetation structure between 36% and 77% by forest restoration compared with degraded ecosystems [34]. Most of the gains were found outside the camp, where long-term care and maintenance were comparatively easier to provide under the jurisdiction of the BFD. The major challenge in sustaining restoration activities within the camp areas was associated with the relocation of refugee settlements inside the camps converting restored areas into other land use. People's high dependency on forest resources (for energy and shelter, etc.) and management initiatives (e.g., species selection, plantation method, collaboration and protection, etc.) along with limited funding and lack of priority for long term maintenance also had direct implications for the success of restoration activities. These factors were more pronounced during the initial stage of the restoration activities. They could be attributed as the driving factors for not attaining the results as expected for restoration activities in 2018. Such landscape context is also identified as one of the major driving factors of restoration success globally [34]. Accordingly, restoration activities in 2019 were more organized, considering the lessons learned from restoration in 2018. A satisfactory sapling survival and growth rate was observed for plantations in 2019 in a field-based plantation assessment [35] which supports the results of this study.

The initial humanitarian responses to land degradation in the area were spontaneous, mainly due to a need for more informed decision-making and collaboration. Over the past years, an integrated approach evolved to reverse the degradation of hundreds of hectares of land. Recognizing the emergency need and the underlying driving factor for land degradation (i.e., land cover changes due to rapid vegetation loss in this case), a simple, intuitive and easy-to-implement method for land degradation assessment, based on evaluation of NDVI dynamics and land cover change, was found effective not only in identifying the degraded land but also in informing the stakeholders about the magnitude and immediate need to respond.

Integrating land tenure information with restoration planning and implementation is critical for achieving restoration outcomes, including land degradation neutrality [36]. In the context of Cox's Bazar, the integration of spatially explicit land degradation information with land tenure (i.e., legal forest land boundary) facilitated the identification and prioritization of land for restoration and other interventions through a geospatial suitability analysis. However, in the early stage of restoration works in 2018, there were instances of scattered plantation interventions with little consideration of the local context. This aggravated the crisis by contributing to conflict. A collaborative, inclusive, evidence-based approach recognizing the direct land users, prevailing complex socioecological system, and land ownership were necessary for success.

Understanding the supply and demand of fuel wood was critical for evidence-based decision-making (e.g., what stocking rates are required for successful reforestation efforts). The assessment of fuelwood supply and demand was necessary, in this regard, to determine sustainable and optimum energy supply from plantations, considering the reduction of demand due to improved energy consumption (e.g., improved cooking arrangements) and/or the use of alternative energy sources (e.g., solar energy and LPG, etc.). Notably, the immense pressure on forest resources revealed by the updated assessment was critical in mobilizing and sensitizing key stakeholders to take immediate action to prevent the complete loss of forest resources. The assessment was also essential for raising awareness of the importance of safe access to fuel and energy.

Preparing technical specifications and guided implementation helped the implementing agencies avoid unplanned activities, protect plant biodiversity and allocate resources efficiently. Stakeholder involvement at every stage of land restoration was essential, requiring substantial coordination between local authorities, community leaders, United Nations agencies, non-governmental organizations and other partner organizations. The collaborative implementation of restoration activities, with due attention to different and sometimes conflicting stakeholders' interests, was critical to the program's success.

Land degradation in an area due to rapid loss of vegetation could be easily detected through remote sensing technologies. Usually, assessing the success of restoration interventions requires long-term records [11], considering the time needed for restoration work to take effect on the ground that can be detected by remote sensing. The time elapsed since restoration began is one of the main drivers of forest restoration success [34]. The restoration works in Cox's Bazar started in less than five years, making it more challenging to assess. In this regard, the approach adopted by comparing the Z scores for the productivity state metric, being sensitive to the recent magnitude and direction of change [24], was found relevant and can be used in a similar context. In general, remote sensing and geospatial analyses were adequate for the preliminary assessment of degradation and the identification and prioritization of suitable restoration areas. However, the results needed to be validated by ground observation before commencing restoration activities. Given the emergency nature of the problem, the approach took a practical and intuitive approach, which was later enhanced with additional data, information and capacity.

## 5. Conclusions

The unprecedented land degradation in Cox's Bazar required an urgent response to restore degraded landscapes. The longer the degradation persisted, the more difficult it

would be to restore the landscape. In the most severely affected areas, there was a substantial risk of irreversible damage, with a total loss of large tracts of forest. There had also been scattered plantation interventions with little consideration of the local context, which aggravated the crisis by contributing to conflict. Implementing and sustaining restoration activities in such an emergency displacement setting was challenging. It required an integrated, multidisciplinary and collaborative approach that considers the entire landscape, including the people and ecosystems it contains.

Within less than five years of restoration work, remote sensing analysis revealed significant positive impacts in most areas brought under restoration activities. The availability of timely information on the status of wood fuel supply and demand, spatially explicit information on land degradation and land tenure, the identification and prioritization of suitable land for restoration interventions, and a collaborative and inclusive approach to implementation were necessary preconditions for such success. Areas where restoration works did not perform as expected were identified along with possible drivers. These driving factors were carefully considered for more effective continuation of the ongoing efforts.

Over the last five years, in the transition from emergency to resiliency in a protracted displacement setting, the approach to address land degradation and manage restoration activities is continuously being updated as more data, methodologies, and capacities become available. Recognizing that every challenge is unique, the integrated approach adopted in Cox's Bazar could be applied–with proper contextualization–in similar displacement settings.

**Author Contributions:** Conceptualization, R.J., R.M., M.T.A.A., S.R., M.F.K. and M.H.; methodology, R.J., R.M., B.A., A.G., P.M.-O., A.V., Y.F., R.d. and M.H.; software, R.J. and A.G.; validation, R.J. and A.G.; formal analysis, R.J. and A.G.; investigation, R.M., M.T.A.A., S.R., M.F.K., M.H.K., M.S.R. and H.N.H.; data curation, R.M., M.T.A.A., S.R., M.F.K. and H.N.H.; writing—original draft preparation, R.J., R.M., M.T.A.A., S.R., M.F.K., A.G., F.M., P.M.-O., A.V., Y.F., G.F., I.J., M.H., B.A. and H.N.H.; writing—review and editing, R.J., R.M., M.T.A.A., S.R., M.F.K., M.D., P.J.A., A.G., F.M., P.M.-O., A.V., Y.F., G.F., R.d., I.J., M.H., B.A., M.H.K., M.S.R. and H.N.H.; visualization, R.J., S.R., A.G. and F.M.; supervision, I.J. and M.H.; project administration, M.D., P.J.A. and M.H.; funding acquisition, M.D., P.J.A. and M.H. All authors have read and agreed to the published version of the manuscript.

**Funding:** This research received no external funding.

**Data Availability Statement:** Not applicable.

**Acknowledgments:** The authors would like to thank the anonymous reviewers for their comments that helped improve the manuscript. The authors are grateful to the Rohingya refugees and host communities in Cox's Bazar, the relevant officials of Arannyak Foundation (AF), Bangladesh Forest Department (BFD), Bangladesh Rural Advancement Committee (BRAC), Center for Natural Resource Studies (CNRS), Food and Agriculture Organization of the United Nations (FAO), Helvitas, International Organization for Migration (IOM), International Union for Conservation of Nature (IUCN), office of Refugee Relief and Repatriation Commissioner (RRRC), Shushilon, United Nations High Commissioner for Refugees (UNHCR), United Nations Children's Fund (UNICEF) and World Food Programme (WFP) for their support during different stages of the work.

**Conflicts of Interest:** The authors declare no conflict of interest.

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
