# Peer review of "Restoring Degraded Landscapes through an Integrated Approach Using Geospatial Technologies in the Context of the Humanitarian Crisis in Cox’s Bazar, Bangladesh"

_land, doi:10.3390/land12020352_

Round 1

Reviewer 1 Report

In this study authors have presents an ongoing geospatial approach that integrates and facilitates different aspects of addressing land degradation in a displacement setting in Cox’s Bazar, Bangladesh. The topic is practical and meaningful. The work done is methodologically correct and achieved the research goal. From my viewpoint, the methods used are appropriate and well chosen for the topic. The work done is well structured and well written with interesting findings. The topic fits the scope of this journal. A few suggestions to further improve the manuscript are given below.

·         Line 44: It is better to define abbreviation ha upon first use.

·         Line 113-114: “majority live in 34, all located in Cox’s” What is 34?

·         Line 191-196: What was the criteria to classify the vegetation into dense sparse vegetation?

·         Line 206-207: What was the source of spatial data such as elephant path and flood risk?

·         Line 134: I suggest authors should add methodological flow chart as Materials and Methods are complex.

·         Figure 5: Please add reference to the Figure and do you have permission to use this map from relevant department/journal/authors?

·         Did you check for normal distribution of data before applying z-statistics?

·         Line 296-297: Please Rewrite. Confusing sentence.

·         Line 214-235: Why it is necessary to discuss these paragraphs here in methodology section? Can you shift it to Introduction, discussion or supplementary materials?    

·         Table 4: What was the criteria to classify the high slope and low slope? Please differentiate.

·         Line 370: The purpose of the discussion is to interpret and describe the significance of your findings in light of what was already known about the research problem being investigated.  Place your study within the context of previous studies. Please rewrite.

·         Journal Name should be abbreviated, Please follow the journals guidelines (https://www.mdpi.com/journal/land/instructions#references)

·         This manuscript contains 24% plagiarism. Your work should not contain any plagiarism.

The above comments justify my recommendation to accept this paper after a 'major' revision. I hope this encourages the authors to improve this interesting study.

Reviewer 2 Report

The present manuscript “Restoring degraded landscapes through an integrated approach using geospatial technologies in the context of the humanitarian crisis in Cox’s Bazar, Bangladesh” presents an interesting study and approach, with scope in the Land, and which will become a reference article and future citations. However, before recommending the present study for publication, some points need correction. Therefore, I am considering this manuscript for minor revisions, highlighting the following points:

1 – The following sentence “The influx of nearly a million-refugee people from Myanmar since August 2017 made Cox's Bazar, the southernmost coastal hill district of Bangladesh, home to one of the largest refugee settlements in the world. This sudden population density increase put significant pressure on the regional landscape resulting in land degradation due to biomass removal to provide shelter and fuel energy and posed critical challenges for both host communities and displaced population.” found in the abstract of the manuscript needs to be summarized in just one short sentence of 2 to 3 lines, as the focus of the abstract is to present the main methodological points and results.

2 – Highlight in the abstract the main methods used in this study.

3 – The text of this article in general needs a complete revision, as there are several spelling and grammatical errors.

4 – The topic “2. Study area and context” can be inserted as a subtopic under “2. Materials and Methods”. Note: both topics have the same numbering, which cannot happen.

5 – Insert the latitude and longitude in Figure 1 (a) and (b). In Figure 1 (c) there is no need for coordinates on the right and bottom of the figure.

6 – Figure 2 must be presented after its first call in the text. Please check the Figure 2 callout in the text.

7 – Insert latitude and longitude in Figure 2.

8 – Line 190 “based on the land cover change from 2017 to 2018 as follows –”, change the “–” to “:”.

9 – Improve the font and quality of Figure 4 (a).

10 – Improve the font and quality of Figure 5, as well as remove the figure's right and base coordinates.

11 – It is fundamental and should be taken as a priority for the authors to add more bibliographic references, as well as to prioritize article references because the current version of the manuscript, is very poor in references and lacks in-depth revisions. I also suggest exploring articles from Land itself!!!

12 – Line 509, does the phrase correspond to reference 21?

Reviewer 3 Report

Dear Authors,

Please, find my report in the attached file!

Kind regards!

Round 2

Reviewer 1 Report

The manuscript was significantly improved after the major revision. 

Reviewer 3 Report

Dear, Authors!

The manuscript was significantly improved after the major revision. However, I have a few more comments and recommendations before the acceptance. 

Line 109 - empty brackets.

Line 130 - Figure 2 depicts...

Line 145 - 2.2. Delineation of forest land boundaries and land cover mapping

Line 153 - real-time kinematic positioning

Line 163 - following the approach recommended by D'Annuzio, R., et al. [21]

Line 386 - ha should be hectares

Line 389 - ...method for land degradation assessment, based on evaluation of NDVI dynamics and land cover change was found...

Lines 397-399 - Please, separate this sentence! It is unclear, and the double usage of initially is inappropriate.

Line 399 - I think "to implementation " is unnecessary here. Just ...approach recognizing...

Best regards!
